# Live births from urine derived cells

P. Olof Olsson[1], Jeong Yeonwoo[2], Kyumi Park[3], Yeong-Min Yoo[4], W. S. Hwang[2]*

**1** Fujairah Genetics Center, Fujairah, UAE, **2** UAE Biotech Research Center, Abu Dhabi, UAE,
**3** Department of Companion Animal & Animal Resources Science, Joongbu University, Geumsan-gun,
Republic of Korea, **4** Lab of Biochemistry and Molecular Biology, College of Veterinary Medicine, Chungbuk
National University, Cheongju, Chungbuk, South Korea

* hwangws@uaebrc.ae

## Abstract

Here we report urine-derived cell (UDC) culture and subsequent use for cloning which
resulted in the successful development of cloned canine pups, which have remained healthy
into adulthood. Bovine UDCs were used in vitro to establish comparative differences
between cell sources. UDCs were chosen as a readily available and noninvasive source for
obtaining cells. We analyzed the viability of cells stored in urine over time and could consis-
tently culture cells which had remained in urine for 48hrs. Cells were shown to be viable and
capable of being transfected with plasmids. Although primarily of epithelial origin, cells were
found from multiple lineages, indicating that they enter the urine from more than one source.
Held in urine, at 4˚C, the majority of cells maintained their membrane integrity for several
days. When compared to *in vitro* fertilization (IVF) derived embryos or those from traditional
SCNT, UDC derived embryos did not differ in total cell number or in the number of DNA
breaks, measured by TUNEL stain. These results indicate that viable cells can be obtained
from multiple species' urine, capable of being used to produce live offspring at a comparable
rate to other cell sources, evidenced by a 25% pregnancy rate and 2 live births with no
losses in the canine UDC cloning trial. This represents a noninvasive means to recover the
breeding capacity of genetically important or infertile animals. Obtaining cells in this way
may provide source material for human and animal studies where cells are utilized.

GERMANY

**Data Availability Statement:** There are no ethical
or legal restrictions on sharing a de-identified data
set, based on the original owner agreements data
may only be made on request. Data may be
requested including all primary blots/gels etc. from

## Introduction

Human and animal cells have several applications, with potential uses ranging from disease
detection and treatments, to reproductive biology, including cloning [1–3]. Cultured cells are
particularly useful for biological applications as they can be used with consistency and
expanded in number for subsequent use. Generally, to obtain cells, a tissue sample is taken, a
process which may be relatively invasive and although it generally does not cause lasting harm,
can lead to complications. There are several risks associated with obtaining biopsies from
sensitive animals, particularly related to potential injury and stress with capture, anesthesia
and infection. Where risks are high or sampling potential is limited, e.g. in critically endan-
gered or particularly sensitive individuals, biopsies may not be a suitable option to obtain cells.
Animals of critically endangered species may not be captured, yet their urine may be obtained.

the UAE Biotech Research Center in Abu Dhabi
(info@uaebrc.ae, +971501878464).

**Funding:** This work was supported in part by the
Korea Institute of Planning and Evaluation for
Technology in Food, Agriculture, Forestry and
Fisheries (IPET), through the Agri-Bio Industry
Technology Development Program, funded by the
Ministry of Agriculture, Food and Rural Affairs
(MAFRA; grant number: 318016-5). The funders
had no role in study design, data collection and
analysis, decision to publish, or preparation of the
manuscript.

**Competing interests:** No competing interests to
declare.

Likewise, in human practice, patients may be hesitant to be subjected to biopsy procedures. Here we show the ability to obtain and culture cells from urine, which may further facilitate cellular-based technologies and research and show that urine derived cells (UDCs) can be used to obtain embryos and live offspring through cloning via somatic cell nuclear transfer (SCNT).

The production of viable offspring, after cloning, validates genomic integrity making clear a range of cellular applications. SCNT produces genetically identical offspring or embryos or in the generation of syngeneic embryonic stem cells (ESCs) [4, 5]. The cell source for SCNT, referred to as donor cells, are generally obtained from biopsies and primary cell culture, most commonly from skin or muscle tissues [6] from living or recently deceased individuals [7]. A significant aspect of cloning efficiency is the ability to reprogram somatic cells. Cell type and passage number have been implicated as effectors of cloning efficiency, with cumulus cells, stem cells and embryonic cells being more efficient in producing offspring, lacking abnormalities, than terminally differentiated cells [8–10]. As cloning efficiencies from terminally differentiated cells, average between 1 to 3% across species, there exists a need to establish superior methods [6, 11]. The ability to produce embryos and live offspring and to produce pluripotent stem cells (iPSCs) formation were considered as hallmarks for cellular integrity and utility. Together these cellular resources and SCNT may prove to be valuable in the restoration of endangered and or extinct species and similar technologies have the potential for human medical advancements [12].

In this manuscript, we describe our investigation into the possibility of using cultured urine-derived cells (UDCs) as donor nuclei for SCNT in canine and bovine species, as well as to show the potential use for transfection in an iPSCs induction system. We additionally aimed to establish the potential for UDCs in conservation related biology and their stability for potential cellular modification as needed for medical research or genetic editing. The canine and bovine species were chosen as the model system due to practicality of access as well as having the cloning systems in place. There are multiple reviews on canine use with many common inheritable diseases [13–15]. The canine genome was sequenced in the early 2000′s [16, 17]. Canine stem cell establishment has been previously described, although several limitations for culture of canine oocytes and embryos persist [18, 19]. Bovine UDCs were selected to generate and compare embryo quality between embryos generated from standard biopsy derived cells, UDCs and *in vitro* fertilized (IVF) embryos as, although problems still exist [20], methods for bovine embryo culture are well established [21]. Additionally, the establishment and use of UDCs from non-related species may indicate the potential to utilize this methodology in multiple species.

## Results

### Deriving cells from urine: Urine-derived cells (UDCs)

Cell cultures, supplemented with sterile filtered urine, were as capable of attachment as their non-supplemented counterparts. An average of $7.1 \times 10^5$ cells were collected per ml of canine urine (n = 26). Bovine urine cells were observed at a much lower concentration, approximately 100-fold less than in canine urine, with an average of $5.3 \times 10^3$ cells per ml (n = 10) (Fig 1). The majority of urine cells were small and round, while larger cells were also present, predominantly in canine samples as they were almost absent from bovine samples. In canines, small cells occurred at an average proportion of 33.68 small to one large cell counted from 21 samples from 14 individuals. Trypan Blue (Sigma) staining was insufficient in determining viability, Hoechst and propidium iodine (P.I.) were used to indicate cell number and nonviable cells, those which had lost membrane integrity. Large cells were never observed to attach and rapidly lost viability. Urine samples held for 24 hrs lacked observable numbers of viable large

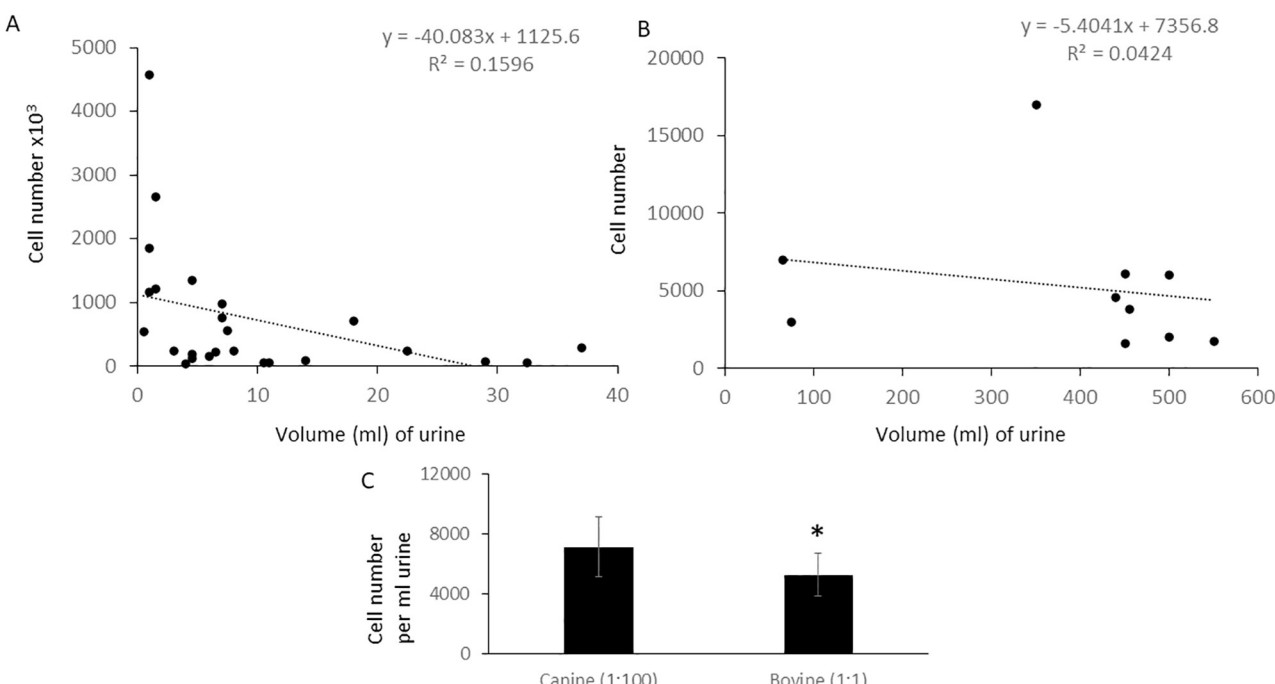

**Fig 1.** Cells number per volume (ml) of urine following micturition collection in A) Canine and B) Bovine samples C) Average cell number per ml, canine numbers displayed divided by 100 for scale. Error bars SEM *(p = 0.036).

cells (Fig 2B). Small cells however, retained their membrane integrity >120 hrs, defined by a negative P.I stain (Fig 2C), although successful adherence (Fig 2D) and derivation of cultivable cells diminished drastically with time in both canine and bovine samples (Fig 2). After 72 hrs, attempts at culturing cells on any substrate, including vitronectin/fibronectin, laminin and collagen yielded only poor results. The percentage of PI negative cells decreased significantly from fresh to 24 hrs (p = 0.014) but did not significantly decrease between subsequent 24 hr time points, after 48 hrs, showing a sharp initial decrease followed by slow degradation of cellular integrity (Fig 2). Eighty percent confluence was achieved at approximately 21 days, in general, however cell seeding density varied based on urine cell concentration. Earliest attachment was seen at 24 hrs and latest observed at three weeks. Delayed attachment generally resulted in failure to achieve sufficient number of UDCs for culture. Cells were found to contain markers for cells of three different origins (Fig 3). The majority of attached cells were of epithelial origin, staining positive for pancytokeratin, although markers for multiple cell types were also present (Fig 3).

Addition of EGF and insulin appeared to increase initial cellular attachment; however cells often receded and appeared to arrest or undergo apoptosis. The loss was corrected with continued dilution, through media changes. Hormonal addition was not necessary to obtain cells, however addition appeared to increase the yield and reduced attachment times, with less failures. 10ng/ml EGF+ Insulin 0.1%, diluted at least 1:1 with the media change/refreshment after 4 days yielded the best and most consistent results. Coating of dishes with either collagen or laminin or the two together, did not increase the frequency or time required for cellular attachment. No addition of insulin, hydrocortisone or growth factors was used in the continued growth of cells after establishment. Original cultures contained an addition of sterile filtered urine, at concentrations up to 20% by volume, without an observed effect on cellular

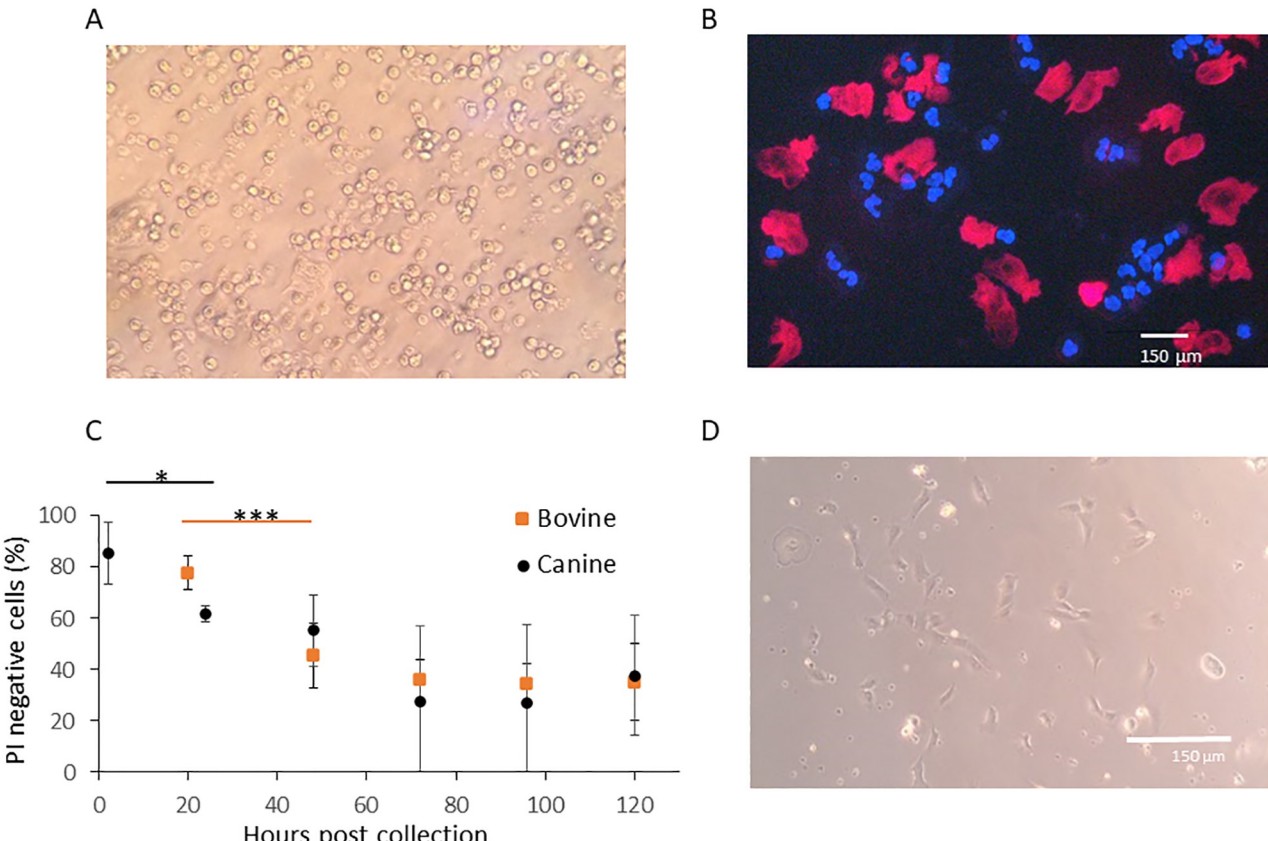

**Fig 2.** Canine Urine and UDCs A) Urine cells following isolation. B) Cellular morphological variation of large and small cells and integrity staining utilizing Propidium Iodide (red) and Hoechst 33342 (blue). Red stained cells indicate membrane permeability and thus non-viable cells; blue stain illustrates nuclei with intact membranes. C) Urine cell integrity indicate by Propidium iodide negative stain over time stored at 4˚C (*p<0.05, ***p<0.001).

adherence or proliferation. When crystals were observed in urine, cellular attachment and proliferation was retarded or ablated. Obtaining UDCs from samples stored more than 48 hrs was less effective, despite the presence of P.I. negative cells (S1 Table).

## Somatic cell nuclear transfer

To determine the developmental capability as donor nuclei for SCNT to produce live offspring, canine fibroblasts and UDCs were allowed to grow to near confluence or confluence before being used for somatic cell nuclear transferF. Fibroblasts were obtained from dermal samples from living donors. Oocytes (n = 87) were checked for fusion and resultant complexes (72.4%, n = 63) were transferred into surrogates (n = 4). Pregnancy was diagnosed and of the four surrogates used, one surrogate was pregnant with 2 pups, consistent with SCNT results using dermal fibroblasts as donor cells (Table 1).

Bovine UDCs generated embryos at rate of 42.0% +/- 17.3SD n = 4 from 250 oocytes, compared to a rate of 26.5% +/- 7.0 SD from SCNT performed with fibroblast donor cells. IVF derived blastocysts formed at a rate of 25.1% +/- 22.1SD. The three group's blastocyst formation rate did not differ significantly. To determine the genetic integrity of bovine embryos derived from UDCs, TUNEL staining was employed and no major differences were observed

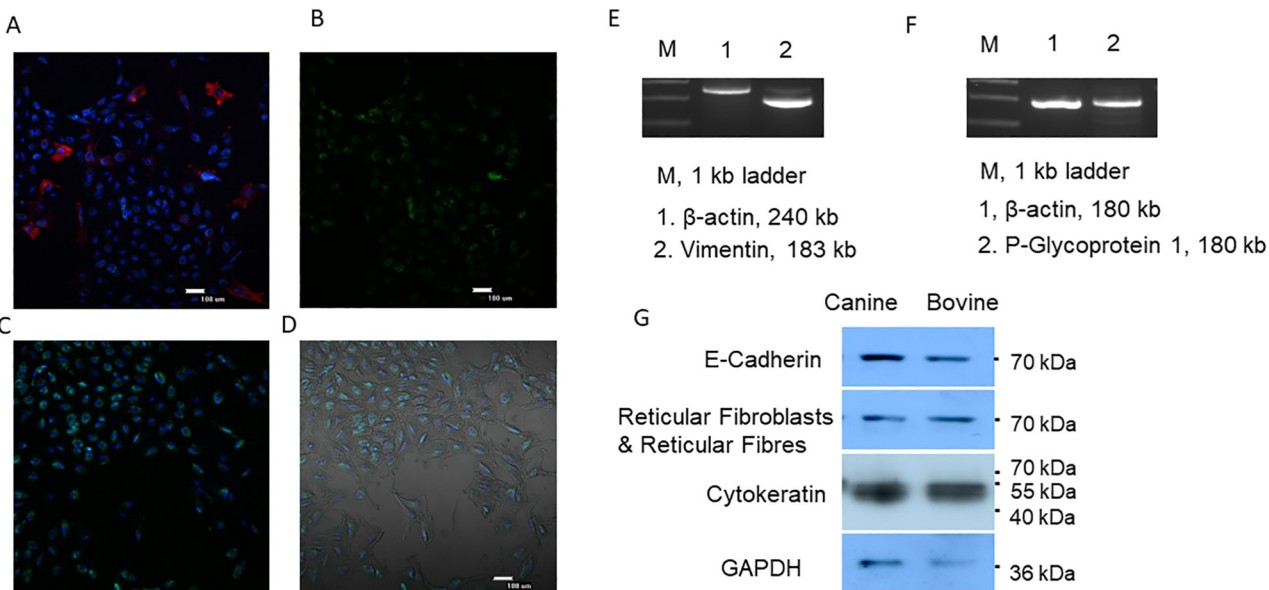

**Fig 3.** UDC immunohistochemical characterization with Hoechst indicating nuclei (Blue) A) Vimentine positive cells (Red) B) E-Cadherin positive staining(Green) C Pancytokeratin stain (Green). D. Pancytokeratin stain overlaid on light microscopy image. E) Presense of Mesynchymal cellular marker Vimentin, F) Epithelial marker P-Glycoprotein. G Further markers for cell characterization indicating a heterogeneous cell population in UDCs.

in SCNT embryos derived from UDCs or fibroblasts, nor from either UDC or fibroblast derived to IVF derived blastocysts. Although the average did total stain differ significantly between groups, there was an increased deviation in the ratio of TUNEL positive cells in SCNT derived embryos from fibroblasts than from UDC derived embryos which in turn was greater than the deviation observed in TUNEL staining of embryos of IVF origin indicating potential biological relevance(Fig 4).

**Production of UDC derived offspring.** After collecting urine, samples were analyzed to determine viability, UDCs were obtained and subjected to SCNT then after embryo transfer, female surrogates were monitored and at day 28 and 30 were submitted to pregnancy detection by ultrasonography. One of the four surrogates was shown to be pregnant and was carrying two pups. Pregnancies were closely monitored and both pups and surrogate were healthy after

**Table 1. Canine cloning comparison and syngeneic cloning attempts utilizing cells obtained from biopsies compared with urine derived cells.**

| Donor cell Transfer | Nuclear transfer | | | | Embryo transfer | | Parturition | |
|---|---|---|---|---|---|---|---|---|
| | No. of oocyte donors | No. of oocytes | | | No. of surrogates | No. of pregnancies[†] | No. of offspring (%)[‡] | |
| | | Retrieved | Subjected to NT | Fused & transferred (%) | | To term (%) | Born | Survived |
| 1st control | 59 | 517 | 502 | 294 (58.6) | 20 | 6 (30.0) | 7 (2.3) | 5 (1.7) |
| 2nd control | 39 | 416 | 407 | 251 (61.7) | 16 | 4 (25.0) | 3 (1.1) | 3 (1.1) |
| Urine derived cells | 10 | 95 | 87 | 63 (72.4)* | 4 | 1 (25.0) | 2 (3.2) | 2 (3.2) |

* Superscript significant difference each column (P<0.05).

1st and 2nd trials and the urine derived cells used were syngeneic.

[†] Pregnancy percentage calculated as the number of pregnancies obtained from the number of surrogates which received reconstructed oocytes.

[‡] No. of offspring (%) represents cloning efficiency, and represents the percentage of live offspring obtained to reconstructed oocytes transferred.

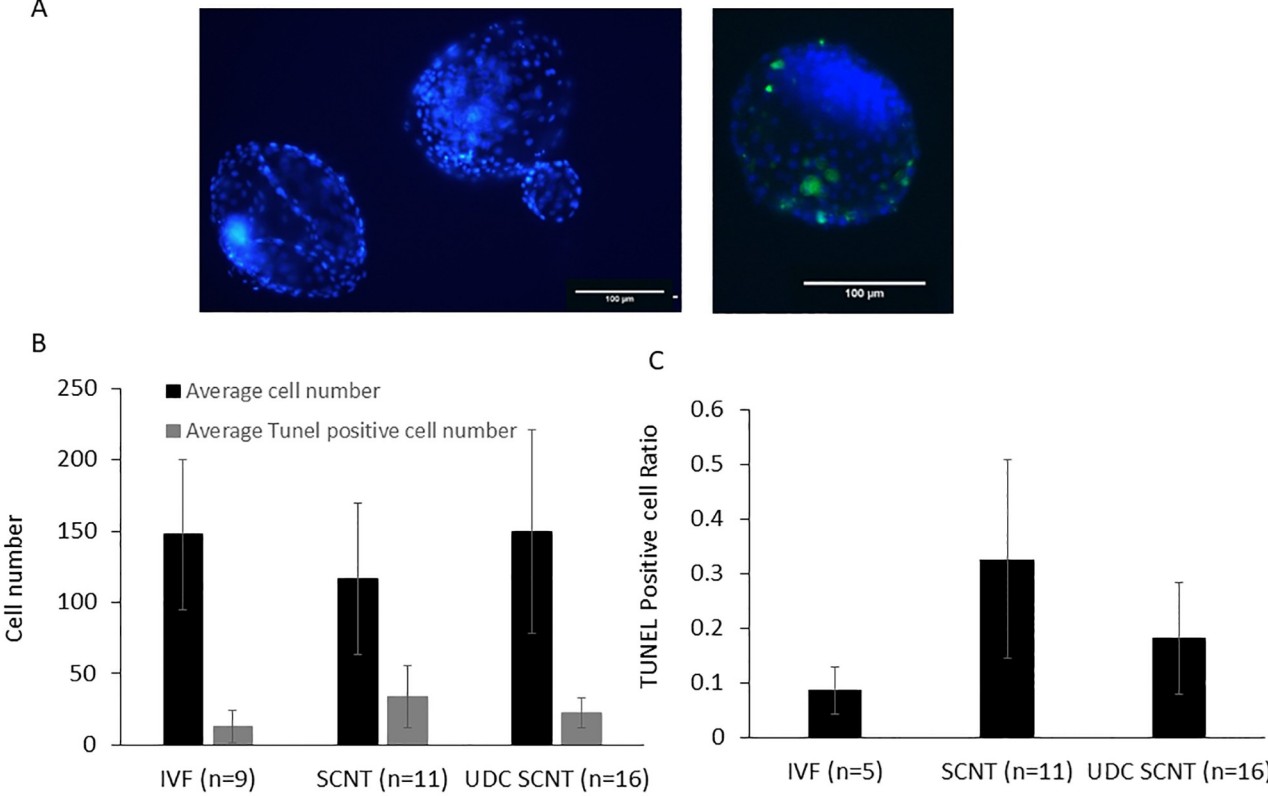

**Fig 4.** Embryo quality analysis A) Representative images of blastocyst with total cells (Blue) and TUNEL positive cells (Green), indicating DNA breaks B) Total cell number and TUNEL staining in 8 day embryos derived from IVF, SCNT with standard grown cells, and UDC SCNT. C) average ratio of tunnel positive cells to total cell counts. Error bars are SD.

parturition (Fig 5). This represents a 25% pregnancy rate and a cloning efficiency rate of 3.2%, with two live births from 87 reconstructed oocytes (Table 1). No abnormalities were detected and pups developed normally with consistent normal growth. When compared with their syngeneic sibling, from the same original donor, and from that of naturally produced pups, growth did not vary (Fig 6A). At the time of submission, the cloned offspring both from UDC and fibroblast origin, have developed normally and passed their fourth year and exhibit no apparent adverse conditions and appear indistinguishable in their health, vigor and vitality (Fig 6).

**Establishment of presumptive induced Pluripotent Stem Cells (iPSCs).** In order to further elucidate the potential of UDCs the pDON-5 plasmid was co-transfected with GFP and used to generate induced pluripotent stem cells (iPSCs). Initial GFP expression could be detected at 24 hrs in culture and initial stem cell-like colonies were visible at 72 hrs, adhesion independent spheroids, reminiscent of embryoid bodies, were observed between 7 and 10 days (S1 Fig). This rationale, that these cells are usefully for multiple purposes, is further supported by the potential applications for iPSCs where cloning is not currently feasible, i.e. in species with inadequate oocyte or surrogate source. Transfection efficiency, evident by GFP expression, was unexpectedly high, 57.34%, +/- 14.69 SD, (n = 3). *OCT4* expression was present when tested 10 days after introducing the plasmid. We did not attempt to optimize culture conditions of iPSCs nor to maintain their pluripotent state or differentiation capacity.

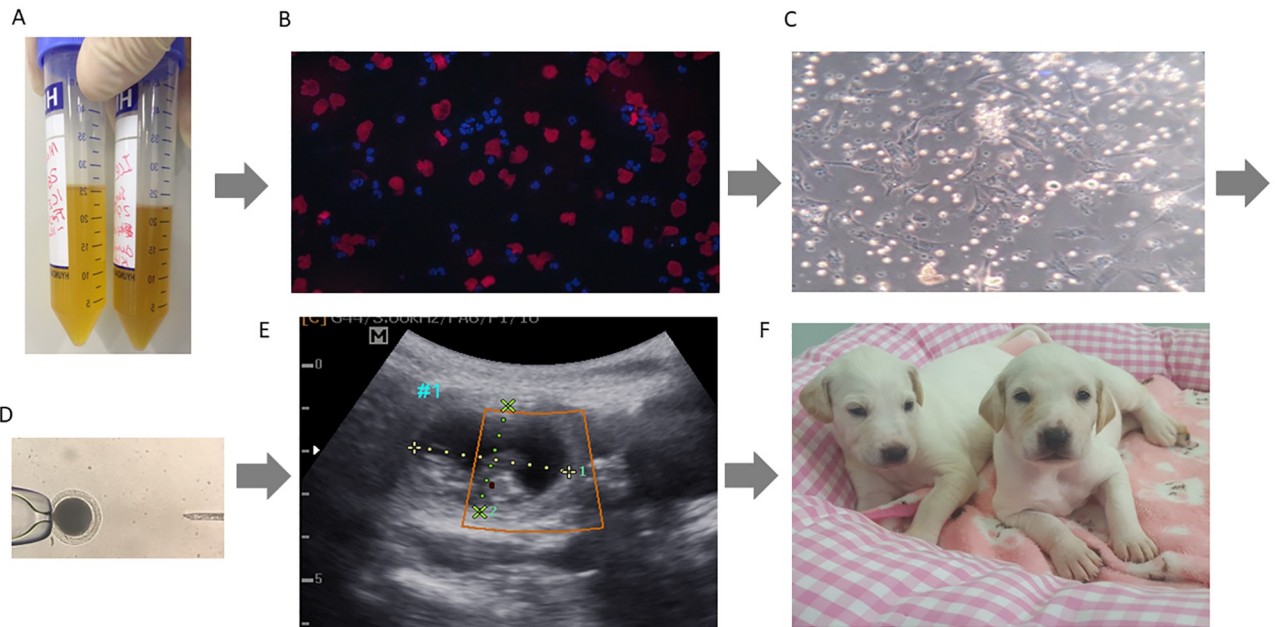

**Fig 5. The UDC cloning flow through.** A) Urine collected, B) Analysis of cellular viability, C) UDCs established D) SCNT performed, E) Ultrasonic pregnancy confirmation at ~30 days after embryo transfer, F) Cloned puppies.

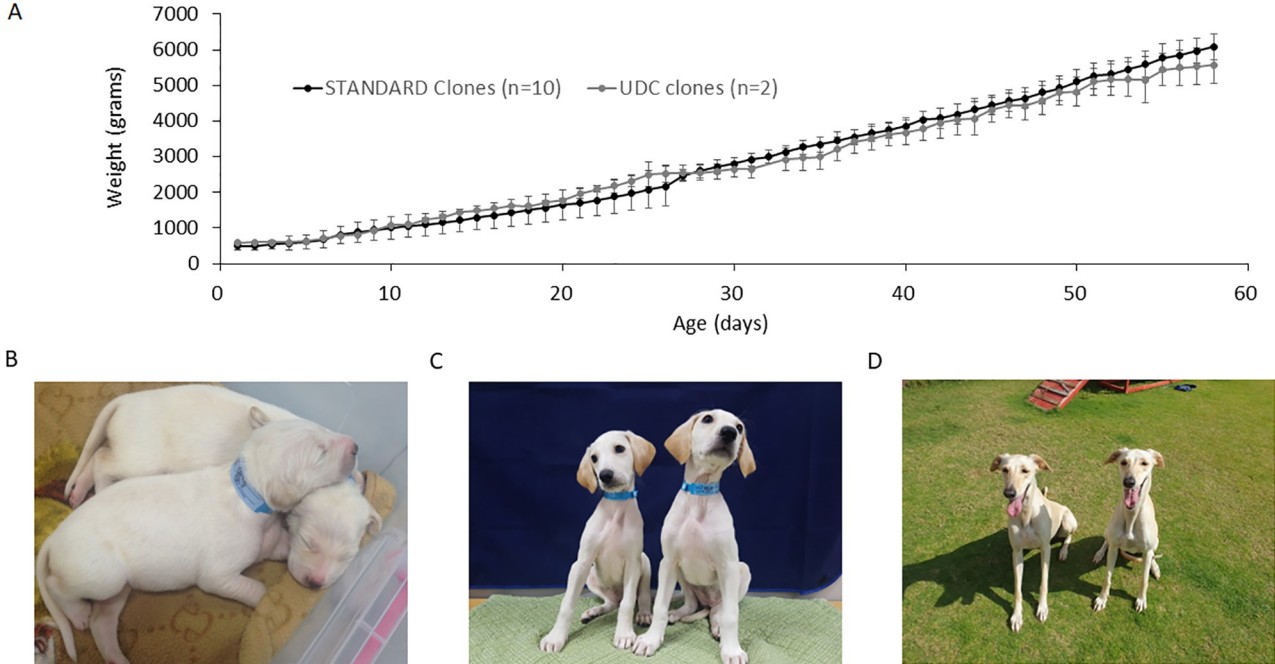

**Fig 6.** Growth and Development of standard SCNT and UDC clone development A) Growth rate until weaning, B) newborns, C) UDC clones at 4 months of age, D) mature UDC clones. Error bars ae SD.

## Discussion

Our results suggest that UDCs are at least as viable a source of nuclear donor material for use in somatic cell nuclear transfer, as conventionally obtained cells, and that they may be useful for other cell-based technologies.

Nuclear transfer for cloning, and for establishing cells and animal and disease models is a long standing technology. This, to our knowledge the first report of a species, other than mice [3] to be cloned from cultured UDCs used as donor nuclei, and the first to use UDCs from clones, or to make the cloning comparisons with other cloned and IVF derived embryos. Fibroblasts are a standard cell type for SCNT, many other cell types have been used and various efficiencies have been reported [22]. This may be in part due to methylation, shown to influence SCNT efficiency [23]. Most biological fluids require some form of ingress to be obtained, urine is an exception and methods for its retrieval may circumvent the need for active sampling. Noninvasive techniques for obtaining somatic cells, including from milk, blood and isolated reports from urine have also been shown, yet few methods have established urine derived cell culture prior to SCNT [24, 25]. Here we show for the first time that canines can be cloned from urine derived cells. Additionally, we have circumstantial evidence, through the presence of what was morphologically an embryoid body (S1 Fig) that iPSCs can be generated from urine directly, as has previously been done with human urine, further validating its potential across species. This is, also to our knowledge, the first report of cloned cattle embryos produced from UDCs and; the only other report of cloning using UDCs was in the production of a buffalo (*Bubalus bubalis*) calf, by means of handmade cloning (HMC) from cells isolated from urine and from milk [26, 27]. In this report they noted significant difference in genes related to histone modifications of H3K27me3 and cellular differentiation and to have a potential regulatory capacity in stem cells [28, 29]. This increased activation potential observed in in UDC derived SCNT may reflect a stem cell population of the donor cell origin [30]. If UDCs have an increased capacity for gene activation, they may make better candidates for gene modification [31]. This may in part explain the unexpected transfection efficiency observed.

In this study, UDCs were generated from a minimum of three different cell and or tissue types (Fig 3) and the urine contained additional cell types which were not represented in the culture or subsequent analysis. This agrees with the reports of up to seven, often difficult to identify, cell types are found in urine [32, 33]. The cells were found to be primarily of epithelial origin but mesenchymal cells were present (Fig 3). These cells may have originated from the bladder, urethra or kidney, likely renal tubular cells [3] or other potential cell types of urothelial origin [34]. Further immunohistochemical staining, i.e. for alpha smooth muscle actin and myosin may provide information to the origin of the VIM positive cells observed [35]. Epithelial cells may originate from the bladder, urethra or kidney, renal and may be proximal tubular epithelial cells CD10/CD13 double-positive [36] Several markers may be used to further clarify the origin of cells within the UDC population. For instance blood, kidney, by the mesynchymal marker CD90 [37], granulocyte/eosinophils by CD16/CD34 and neutrofiles CD11b/CD18 [38, 39], pericytes by NG2 [40] and myofibroblast by aSMA [41]. The epithelial cells observed may further be identified by markers such as Cytokeratin 7, indicating epithelia lining cavities of the internal organs, ducts and vessels or CD45 positivity for hematopoietic cells [42]. Further investigation is warranted to determine the cell source and results may be inferred from the study of other species. Human and murine studies, combined with the potential for medical revelation, have a more complete molecular toolset available, and may be better candidates for study.

It has been suggested that urinary epithelial cells may be a suitable source for transplantation utilizing stem cell technologies [43]. The population and origin of UDCs is not completely

understood, although we have indicated there are likely multiple originators including a potential stem cell component, and thus more pluripotent cell state may have contributed to the success in the use of UDCs for SCNT.

Cell number and culture efficacy varied widely between individuals and urinations. In general, when crystals were observed in urine, efficiency decreased, although there was too little consistency to objectively determine the differences. Health, age and hydration are likely variables which may contribute to the number and quality of cells, culturable or otherwise in urine. In canines, it was empirically observed that darker urine, presumably due to a dehydrated stated, contained fewer cells, those present appeared highly keratinized and culture efficiency was poor. UDCs were obtained from animals of varying ages and no observable differences were seen between breeds in either canine or bovine groupings. The significant difference in the number of cells per volume of urine between canine and bovine specimens with an approximate 100-fold difference by volume remains unexplained (Fig 1). Urination volume did not strongly correlate with cell numbers, a potential indication that the majority of urine cells may originate from epithelial linings, pulled away during the flow of urine (Fig 2). This would be consistent with the apparent majority of cells being epithelial in nature (Fig 3). The large cells observed were not identified, they could represent a number of cell types reported in urine from macrophages to lymphocytes and eosinophils [32], but further species specific investigation would be required.

Despite the prolonged exposure to urine both in the body and in storage, up to 140 hrs many cells maintained their membrane integrity, and some were able to be cultured, up to 72 hrs (Fig 2). Due to the negative P.I. stain, we theorize that these cells may still be usefully in either direct SCNT or use in obtaining iPSCs, further study is required. There is one report of mice being cloned directly from urine cells [3], the viable uncultured cells may prove functional in this application, or for the generation of ESCs. It was a surprise that cells maintained their viability after prolonged storage of urine at 4°C, initial thoughts were that may be related to the presence of the function urate oxidase (UOX), a gene in canids and other mammals, lost in hominids, its product, uricase, is involved in urate degradation, which is toxic to cells, to allantoin [44, 45].

Some practical limitations to urine collection and transportation exist and collection of wild animal urine may be limited to species with routine urination or marking behaviors. Collection from retromingent animals is more easily accomplished, yet contamination remains an issue as aseptic collection of any animal urine is difficult. We were able to culture cells collected from urine obtained from a cleaned surface, however multiple attempts and several washing and centrifugation steps were required before cells could be obtained that were free from contamination. A limitation and area for further investigation would be a comparison of UDC iPSC induction efficiency, as we did not have a direct comparison with fibroblasts. Additionally, direct induction of fresh urine cells, which has been reported in other species, further justifies the continued investigation of urine as a cell source [46].

One disadvantage to this method is the relatively low number of initial cells obtained and the time required to obtain confluence, approximately 21 days in most cases. Efficiency of culture can be increased using growth factor and cytokine supplementation, with i.e. Single-Quots (Lonza). Due to this low relative growth, we wondered about senescence, which is known in a number of disorders affecting the urinary system [47]. A benefit to the use of this method for SCNT is the low number of cells required, theoretically one cell per oocyte. In practice, larger cell numbers are easier to work with and cloning efficiencies are increased by confluent culture by contact inhibition and culture methods [48]. The 25% pregnancy rate and cloning efficiency achieved (Table 1) in the production of pups from UDCs was consistent with cloning rates we have observed and those reported by others [49]. Although SCNT

rates are reported to be quite low, pregnancy rates can be significantly higher, depending on the species, number of oocytes and method of transfer [50]. Variables in efficiency rates can be influenced by a number of factors including: donor cell type, methylation status, cell cycle stage, transfer techniques and oocyte quality 11. Varying cloning efficiency rates been reported, with the vast majority under 5% [11, 51], this warrants further investigation for cell type and led to the testing of iPSC generation, from which embryoid body like structures were formed (S1 Fig).

The transfection rate following iPSC plasmid transfection was unexpectedly high. The resultant cell clusters exhibiting adhesion independent growth, resembled embryoid bodies. Further investigation to the potential growth, culture and developmental competence of these cells would be required to determine the nature of their pluripotency. Recent work has also established the potential of complete male and female gamete generation *in vitro* in multiple species, including from stem cells [52–55]. Furthermore, mice have been produced with the genetic material from cells originating from two male mice [56]. Several of our associated labs have expressed difficulty in the establishment and maintenance in canines, it wasn't until recently that canine iPSCs were produced [57]. Although ES and iPSCs from canines have been established, standard protocols for their culture are currently lacking [58]. The transfection results and presumptive embryoid body formation we observed would indicate that canine iPSCs may not be as difficult to produce and culture as we had assumed. It is postulated that certain cells are more capable of being reprogramed, although other experiments and our own observations have shown that difficult to determine [59]. This difference in cloning efficiency and phenotypic variation remains poorly characterized and without explanation, although generally assumed to be due to epigenetic reprograming failure [8, 60]. This warrants further investigation of cellular reprograming and iPSC induction prior to use in SCNT, which may increase cloning efficiencies which would in turn decrease or increase the potential positive outcome of difficult cloning cases.

We attempted to determine the maximum storage time of urine to obtain viable cells. We were able to successfully culture UDCs from urine stored up to 72 hrs, although the efficiency dropped after 48 hrs and obtained membrane integrity for more than 120hrs, indicated by negative PI staining. This further validates the potential use of urine as cell source for conservation biology, addressing concerns about storage times associated with urine obtained in remote places.

## Conclusion

In this study we have verified the potential for UDCs as a noninvasive source for obtaining cells which could be used for producing viable offspring and likely a number of applications including diagnostics and or human therapeutic uses [43]. The methods warrant further investigation and other applications may be arise from our production of live offspring, embryo analysis and UDC transfection and potential iPSCs formation [61].

We conclude that urine is both a viable cells source of vital cells and that UDCs may be obtained from multiple species, that UDCs are useful for expansion as donor nucleus for somatic cell nuclear transfer and dedifferentiated stem cells capable of producing viable progeny.

## Methods

All materials were purchased from Sigma-Aldrich, unless otherwise indicated.

## Animals

Urine was obtained from adult and adolescent, male and female dogs, analysis was conducted on all cell types. Urine-derived cells used for cloning were done so from clones, which acted as their traditional biopsy controls. Surrogate and donors were used from mixed breed dogs with ages between 2 and 7 years (body weight 20-25kg), housed in indoor kennels, fed standard commercial dog food once daily, and given water *ad libitum*. All animals included within this study were done so with owner consent. All animal procedures were conducted in accordance with the animal study guidelines and approved by the committee at the Sooam Biotech Research Foundation, Korea (permit no. C-12-01).

## UDCs

Urine was obtained from bovine (n = 27) and canine (n = 10) origin, from multiple breeds, sampled during micturition in a sterile vessel, or collected from a sterilized surface and placed directly on ice or in refrigeration at 4 degrees centigrade until further use. Samples were stored for various time points, but used within a 1hour +/- variation for all time points given. Urine was centrifuged at 150 RCF and washed with phosphate buffered saline five times before resuspension in DMEM+20%FBS, counted and diluted to $1x10^5$ cells/ml as available. Urine cells were cultured in media containing 10–20% sterile filtered urine (0.2μm filter), from the supernatant of the first centrifugation + a mixture of hormones and substrates including 0.5 μg/mL hydrocortisone, 10ng/ml bFGF and 0.1% insulin. Culture plating with protein: BSA (1%), Laminin, Collagen, Vitronectin and Fibronectin were done with 0.12-1mg/ml in 0.4ml 4 well dishes (Cat No 176740 Nunc, Thermo Fisher Scientific) for 1hr at 37˚C were washed gently 1x with PBS before seeding with urine cells (S1 Table). Cells from fresh urine consistently adhered to culture plates by day six. Increased contamination rates and an inability for cells to attach to culture plates increased with storage time, cells stored in chilled media or from freshly collected urine fared better.

**Preparation of cells for SCNT.** To obtain cells for nuclear transfer; a skin biopsy was obtained from a living Saluki. These clones would later be used as a syngeneic control for UDC SCNT. After mechanical and enzymatic dissociation of the tissue (Hossein, 2009 #2), cells were allowed to attach to a culture dish (Product No. 353002, Falcon®, Corning) in Dulbecco modified Eagle medium (Cat. #11995065, DMEM high glucose pyruvate, GibcoTM) with 10% fetal bovine serum (Cat. #16000044, FBS, GibcoTM) at 37 ˚C in an atmosphere of 5% CO2 and air. Media was changed every 48 hrs until the cellular monolayer reached approximately 80% confluence. UDCs used in the cloning experiments were obtained, as described above (UDCs), from two cloned saluki pups of the same donor origin. Both explants and UDCs were cultured until they approached 90% confluence, trypsinized and reconstituted at concentrations of approximately $1 x 10^6$ cells per mL, then cryopreserved in cryovials containing DMEM + 20% FBS +10% DMSO. Prior to cryopreservation cells were taken in the same manner to be used as donor cells for SCNT. After cell populations reached near confluence, cells were used for SCNT (Nuclear Transfer).

**Nuclear transfer (SCNT).** Canine oocytes were collected from domestic canines exhibiting spontaneous estrus, with ovulation status determined by serum progesterone concentration, as previously described [62]. Oviducts were flushed bilaterally with 10ml TCM 199 supplemented with HEPES (Invitrogen Corporation, Carlsbad, CA). Oocytes were enucleated after cumulus cell denuding and pre-staining with 5μg/ml bisbenzimide (Hoechst 33342) as previously described [63] utilizing an inverted microscope equipped with epifluorescence (TE2000-E; Nikon Corporation, Japan).

Bovine oocytes were obtained from aspiration of ovaries obtained from a local abattoir, oocyte complexes were cultured in groups of 20 to 25 per well in a 6-well dish for 40 to 42 hrs in BO-IVM (IVF Bioscience, Falmouth, UK) at 38°C in 5% CO2 in humidified atmosphere after they had first been selected for homogenous cytoplasm with adequate cumulus layers and washed three times in Dulbecco's phosphate buffered saline (DPBS; Welgene, Gyeongsan, KR) supplemented with 5 mg/ml bovine serum albumin (Thermo Fisher Scientific, Waltham, MA, USA) and 1% (v/v) antibiotic-antimycotic (Thermo Fisher Scientific, Waltham, MA, USA). For in vitro maturation (IVM), selected COCs were cultured in Enucleated metaphase II oocytes and donor cells, a dermal fibroblast or UDC derived cell, with a smooth surface, was transferred into the perivitelline space of an enucleated canine or bovine oocyte, in the same manner, using a fine glass pipette. Cell oocyte couplets were equilibrated in a 0.26 M mannitol solution containing 0.5 mM of HEPES, 0.1 mM of $CaCl_2$ and $MgSO_4$ for 4 min. Following equilibration couplets were transferred to a chamber containing mannitol, with two platinum electrodes. Couplets were fused using two DC pulses of 1.75 to 1.85 kV/cm for 15 μs, provided by a BTX Electro-Cell Manipulator 2001 (BTX, Inc., San Diego, CA, USA). After simultaneous fusion and activation, reconstructed oocytes, were cultured in groups of 5 to 6 in 25 μL micro-drops of mSOF covered with mineral oil for a maximum of 1 hr at 39 °C in a humidified atmosphere (5% $O_2$, 5% $CO_2$ and 90% $N_2$) until embryo transfer [64]. Bovine couplets were fused in a similar manner with two 580 Volts, 15 mS pulses reconstructed embryos were activated by treatment to 5 μM ionomycin for 3 min and subsequently with 2 mM 6-dimethylaminopurine (6-DMAP) in BO-IVC (IVF Bioscience, Falmouth, UK) under a humidified atmosphere of 5% $CO_2$ at 39°C for 4 hrs. Embryos were then cultured at 38°C in a humidified atmosphere of 5% $CO_2$ and 5% $O_2$ in groups of 6 to 8 per droplet oil-covered for 7–8 days at which point they were fixed in 0.2% Formalin before embryo evaluation.

## Collection of oocytes, embryo transfer and pregnancy diagnosis

Surrogate dogs exhibiting natural estrus were matched with oocyte donors at the same cycle stage as described [64]. The ovary exposed via a ventral laparotomy, the fat layer covering the ovary grasped with forceps and suspended with a suture to exteriorize the fimbriated end of the oviduct. Reconstructed embryos were loaded into a tomcat catheter (3.5 Fr × 5.5"; Severeign, Sherwood, USA) with a minimum volume of fluid (2 to 4μl) and gently transferred into the 2/3 distal position of oviduct through infundibulum. After 25 to 30 days, pregnancy was confirmed using transabdominal ultrasound with a real-time ultrasonography. Ultrasonography was performed either in standing or in a dorsal recumbent position using a portable ultrasound machine with a 3.5 MHz curved transducer (Sonace R7; Samsung Medison, Seoul, Korea). No bovine embryos were transferred.

## Transfection using iPSC plasmids

Urine cells washed and held in PBS as well as UDCs, trypsinized and held in PBS were electroporated in the presence of our plasmid then cultured in ES medium atop a bed of mitomycin C inactivated mouse embryo fibroblasts (MEFs) as described [65] for 10 days. Cells were imaged using a fluorescent microscope (Nikon, Eclipse TE2000).

iPSC generation was done using the pDON-5 OSKLN plasmid co-transfected with a generic GFP plasmid and the iPSC inducing plasmid pDon-5 OKSLN (Cat No. 3671 Takara-bio, Shiga, Japan). The plasmid was linearized prior to use using the restriction enzyme DraI at 5μg of plasmid in a reaction volume of 30μl. Plasmids were purified using equal volume of phenol chlorophorm followed by isopropanol precipitation and washing before being suspended in 10μl of distilled and autoclaved water. 1μg of plasmid per ml was used for transfection with a

single 20ms pulse of 1600V. Transfection efficiency was validated using a circular GFP plasmid. Cells expressing GFP were considered to be positively transfected and were further confirmed with staining for OCT3/4 and CDX2. Cells were plated in ES media (DMEM/F12) with 1% NEAA, 1% glutamax, 1% antibiotic antimycotic, 0.1% beta mercaptoethanol, 1% B-27 Serum Free supplement 50x, 1% N-2 supplement, 3μM CHIR99021, 25μM PD98059 (Cat NoPHZ1164 Thermo Fisher Scientific), 10ng/ml bFGF recombinant human protein and 20% KnockOutTM Serum replacement. Following verification of transfection and morphological change, further culture was not attempted.

## Polymerase Chain Reaction (PCR)

Cells were treated with TRIzol (Ambion, Austin, TX, USA). Total RNA concentration was determined at absorbance 260 nm. 1 μg of RNA was transcribed by MMLV (iNtRON Bio, Gyeonggi-do, Republic of Korea) with a 9-mer random primer (TaKaRa Bio, Kusatsu, Japan) to produce the complementary DNA (cDNA). PCR was performed with 10 pM of each specific primer. The primer sequences were as follows: permeability-glycoprotein 1 (P-gp) or multidrug resistance protein 1 (ABCB1), forward ctcgcatcttgcttctggat, reverse gctccttgattctgccattc; β-actin, forward ctcttccagccttccttcct, reverse gggcagtgatctctttctgc. And for Vimentin: forward Vimentin, reverse ttcgacggcaaagttctctt; β-actin, forward agaacatcatccctgcttc, reverse ttgaagtca-catgagaccac. The PCR was performed 40 cycles under the following conditions: denaturation at 95˚C for 30 sec, annealing at 59˚C for 30 sec, and extension at 72˚C for 30 sec.

## Immunohistochemistry and staining

The cell numbers of blastocysts were counted following HOESCT staining and imagine under a fluorescent microscope. Immunohistochemical stains and the proportion of cells stained were counted similarly using ImageJ estimated ICM and Trofectoderm cells were not distinguished and total blastocyst cell count was used as a measure of developmental quality at day 6–8 post SCNT or IVM, in the case of controls. Embryos were fixes and stained for TUNEL, TUNEL cells were counted and compared between groups. Nearly confluent cells cultured from direct urine derivation or sub-cultured frozen cells were washed with PBS and fixed in 4% formaldehyde, washed with PBS 1% BSA and 0.1% Tween and blocked with the same washing solution+ 10% donkey serum for 6 hrs at room temperature. Primary antibodies for E-Cad, VIM, and pan-cytokeratin were used at a dilution of 1:500, in 4% Donkey Serum in PBS supplemented with 1%BSA and 0.1% tween at 4C for 8 hrs. (overnight).

Stained cells were washed 4x with PBS container 1%BSA and 0.1% tween and were counter stained with no secondary antibodies or Donkey anti-Rat, anti-rabbit, and anti-mouse. 488, 488 and 555 respectively. Secondary antibody dilution of 1:500 was done in PBS + 1% donkey serum. Staining was done in the culture with cells which had been cultured on glass coverslips added prior to seeding.

Hoechst was added and allowed to stain for 5 minutes at the second to the last of 5 PBS washes prior to imaging. Images were taken on an inverted fluorescent microscope (Nikon, Eclipse TE2000) and analyzed using ImageJ.

## Western blot

Western blot analyses were performed as previously described unless stated otherwise [66]. $1 \times 10^5$ cumulus cells were sampled and lysed using the EzRIPA Lysis kit (WSE-7420; ATTO Corporation, Republic of Korea) following the manufacturer's instructions. BCA (Bicinchoninic acid) $^{TM}$ Protein Assay kit (23225; Thermo Fisher Scientific) were used to for protein determination, and 10 μg samples were separated on 12% SDS-PAGE gels prior to transfer to

polyvinylidene fluoride (PVDF) membranes. PVDF membranes were incubated at 4°C with primary antibodies: Anti-E Cadherin (DECMA-1) and Anti-Reticular Fibroblast and Fibers (ER-TR7)(Abcam) and Vimentin and Cytokeratin Pan Type I/II (MA5-13156) (Thermo Fisher) at 1:1000 dilutions overnight. After overnight incubation, the membranes were incubated with the appropriate IgG-conjugated horseradish peroxidase secondary antibody (Cell Signaling Technology) for 2 hrs at room temperature. Immunoreactive proteins were visualized using X-ray film and the bands were imaged using a scanner. ImageJ software (version 1.37; Wayne Rasband, NIH, USA) was used for the measurement of optical density and the data were normalized to internal control. All experiments were repeated three times (n = 3).

## Statistical analysis

Offspring development rates were evaluated using chi-squared tests. $P < 0.05$ was considered significant. A one-way ANOVA was used to evaluate differences between TUNEL staining for the three conditions and students t-tests were used for comparison between two groupings, such as cell number and viability between two groups.

## Supporting information

**S1 Fig.** A) co-transfected canine UDCs with an iPSC inducing plasmid and expressing GFP (green) and UDCs morphological change 48hrs following transfection. B) embryoid body like formation C) senescent cell morphology and embryoid body like structure 10 days following iPSC plasmid transfection.
(TIF)

**S1 Table. UDC culture conditions and result of culture.** Effectivity compared with the same conditions, replicates of 3 or more were done in each case using the same urine source and cell number, effectivity represents observational differences alone rated "+" beneficial, "n" = neutral or effect not noticeably different than the control and "-" representing negative or potentially negative effects. *-/+ concentration dependent, high concentration negative effect, lower concentration positive effect.
(DOCX)

## Acknowledgments

We would like to acknowledge Bae JinHyun for her assistance in urine collection and Mr. Han and Dr. Lee for their support. We would like to additionally acknowledge His highness Mohammed bin Rashid Al Maktoum for the ability to obtain urine and produce and report on the UDC clones from his dogs.

## Author Contributions

**Conceptualization:** P. Olof Olsson, Jeong Yeonwoo, Kyumi Park.

**Data curation:** P. Olof Olsson, Jeong Yeonwoo, Kyumi Park.

**Formal analysis:** P. Olof Olsson.

**Funding acquisition:** W. S. Hwang.

**Investigation:** P. Olof Olsson.

**Methodology:** P. Olof Olsson, Jeong Yeonwoo, Yeong-Min Yoo.

**Project administration:** P. Olof Olsson, W. S. Hwang.

**Resources:** P. Olof Olsson, W. S. Hwang.

**Supervision:** P. Olof Olsson, W. S. Hwang.

**Visualization:** P. Olof Olsson.

**Writing – original draft:** P. Olof Olsson.

**Writing – review & editing:** P. Olof Olsson.

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
