## [Decision Letter · Decision Letter 0]

22 Apr 2022

PONE-D-22-06002Dogs birthed from urine: A noninvasive source of cells for reproductive cloning and other technologiesPLOS ONE

Dear Dr. Olsson,

Thank you for submitting your manuscript to PLOS ONE. After careful consideration, we feel that it has merit but does not fully meet PLOS ONE’s publication criteria as it currently stands. Therefore, we invite you to submit a revised version of the manuscript that addresses the points raised during the review process.

For quite sometime, I was unable to secure a second reviewer who would accept the reviewing duty.  Therefore I served as the second reviewer.  After making comments on just a few pages, I realized that without page/line numbers, the task of review is very confusing.  I therefore did not finish the first round of review.  Please address comments from Reviewer 1 and my limited comments from the first few pages.  I will be able to finish the first round of review after page/line numbers are added.  Please see my comments below: 

There are no line or page numbers.  Need to add them.

Abstract: cells can be obtained from genetically important or infertile animals without using urine.  The use of urine cell is not well justified.  The statement needs re-written.

Remove UCD and UDC, there are many inconsistencies and are confusing, just use urine cells.

Introduction: references are preceded with sentence stop “.”.  the stop should be placed after the references.  References separated from the preceding word with a space. This should be changed throughout the manuscript.  

What are sensitive animals?

For cell sampling, the authors only discussed tissue biopsy, blood samples are not discussed.

“Cell type and passage” the sentence needs re-written.

The sentence “To determine” needs to be re-written.

Mitochondria from donor cells have been shown to be present in cloned offspring. Citing one paper that did not find them is not appropriate here.

Remove “employed”.

We look forward to receiving your revised manuscript.

Kind regards,

Xiuchun Tian

Academic Editor

PLOS ONE

Journal Requirements:

“This work was supported in part by the Korea Institute of Planning and Evaluation for Technology in Food, Agriculture, Forestry and Fisheries (IPET), through the Agri-Bio Industry Technology Development Program, funded by the Ministry of Agriculture, Food and Rural Affairs (MAFRA; grant number: 318016-5).”

5. Please include a copy of Table 2 which you refer to in your text on page 7.

7. PLOS ONE now requires that authors provide the original uncropped and unadjusted images underlying all blot or gel results reported in a submission’s figures or Supporting Information files. This policy and the journal’s other requirements for blot/gel reporting and figure preparation are described in detail at https://journals.plos.org/plosone/s/figures#loc-blot-and-gel-reporting-requirements and https://journals.plos.org/plosone/s/figures#loc-preparing-figures-from-image-files. When you submit your revised manuscript, please ensure that your figures adhere fully to these guidelines and provide the original underlying images for all blot or gel data reported in your submission. See the following link for instructions on providing the original image data: https://journals.plos.org/plosone/s/figures#loc-original-images-for-blots-and-gels.

Reviewers' comments:

Reviewer's Responses to Questions

**Comments to the Author**

1. Is the manuscript technically sound, and do the data support the conclusions?

Reviewer #1: Yes

2. Has the statistical analysis been performed appropriately and rigorously? 

Reviewer #1: Yes

3. Have the authors made all data underlying the findings in their manuscript fully available?

Reviewer #1: Yes

4. Is the manuscript presented in an intelligible fashion and written in standard English?

Reviewer #1: No

5. Review Comments to the Author

Reviewer #1: The manuscript describes the use of urine derived cells (UCD) for canine somatic cell nuclear transfer. They demonstrate the ability to culture cells isolated from urine, generate cloned embryos and produced cloned puppies (the authors have established expertise in canine cloning). The authors were also able to generate cloned embryos from bovine urine derived cells. The science is interesting and sound and has solid application for a non-invasive source of donor cells for SCNT. The manuscript does need considerable editing for sentence clarity/structure and punctuation (mainly comma usage) to make it suitable for publication. Especially in the abstract where it needs to be clear the work that was done in canine versus bovine. This reviewer typically provides edits to assist the authors, but without page numbers or line numbers, that task is particularly onerous. The discussion is overly long and the discussion of the H3K27me3 in buffalo clones (ref 27) doesn’t add much, since there is no epigenetic analysis in this paper. Since the fibroblast and UCD donor cells were obtained from cloned animals, the potential impact of serial cloning on efficiency should be discussed. Table 2 is missing from manuscript.

6. PLOS authors have the option to publish the peer review history of their article (what does this mean?). If published, this will include your full peer review and any attached files.

Reviewer #1: No

---

## [Author Response · Author response to Decision Letter 0]

5 Jul 2022

The response to all review questions is attached and all items identified have been addressed.

---

## [Decision Letter · Decision Letter 1]

22 Aug 2022

PONE-D-22-06002R1Live births from urine: A noninvasive source of cells for reproductive cloning and other technologiesPLOS ONE

Dear Dr. Olsson,

Thank you for submitting your manuscript to PLOS ONE. After careful consideration, we feel that it has merit but does not fully meet PLOS ONE’s publication criteria as it currently stands. Therefore, we invite you to submit a revised version of the manuscript that addresses the points raised during the review process.  

The revised version is still in very poorly prepared form.  The manuscript is full of inconsistencies and errors such as abbreviations (hour, hrs), stain names, bold/fonts of section headings, expression of exponentials, space missing between numbers and their units, before brackets, missing part of a figure legend, missing punctuations. A thorough professional editing on English and writing is REQUIRED. Below, I listed a number of changes that need to be made.  These are by no means the only problems.  The sub-standard preparation prevented me from going through the entire manuscript due to the sheer number of English and composition issues.  It is the job of the professional editing service and the authors to ensure that the writing is at the level scientific publication before submission.   

Abstract: there was no mention of birth and pregnancy, and too much on cells and culture.

Remove UDC throughout the manuscript as required from the first round of review.

Introduction:

L42: replace one of the “range” with another word.

L43: “modeling” is not the right term here.

L52: remove “to obtain a tissue sample”

L53-54: remove one of the “further”

Please add in Introduction: animals of very endangered species may not be captured yet their urine may be obtained. 

L56 and 57: needs re-written. The two issues before and after “and” are not equivalent.  The second one is particularly unrelated to “the production”.

L61: add a space before [7].

L66-68: sentence grammatically incorrect.

L68-71: as it was pointed out already in the first round of review, mtDNA from donor cells has been found to remain in the SCNT embryos and progenies.  Please change and add more references.

L71-73: please remove one of the “tool”.  Also, what are “these tools”?  The preceding statement was about mitochondria.  In this entire manuscript, a common problem in the writing is that the sentences jumped around in sequences.  There is widespread lack of logical progression or connection. 

Results:

L92: Please mention the supplements. 

L93: please change “+/-“ to its publishable format throughout the manuscript.  Remove “S.D”.

L96: what are the criteria of “small” and “large”?

L98-99: any number below 10 should be spelt out. 

L100: please explain why you determined that Trypan Blue did not work for viability.

Discussion at more than 7.5 pages is too long, please shorten by half.

L357: did you have 27 bovine or 10 bovine?

Table 1: under the column heading “transfer method”, you listed “1^st^ trial”, this is not a method.

Fig. 1.  The letter “C” is embedded in B, figure legend has grammatical errors, lack of proper labeling for both X- and Y-axes in C.

We look forward to receiving your revised manuscript.

Kind regards,

Xiuchun Tian

Academic Editor

PLOS ONE

Reviewers' comments:

Reviewer's Responses to Questions

**Comments to the Author**

1. If the authors have adequately addressed your comments raised in a previous round of review and you feel that this manuscript is now acceptable for publication, you may indicate that here to bypass the “Comments to the Author” section, enter your conflict of interest statement in the “Confidential to Editor” section, and submit your "Accept" recommendation.

Reviewer #1: (No Response)

2. Is the manuscript technically sound, and do the data support the conclusions?

Reviewer #1: Yes

3. Has the statistical analysis been performed appropriately and rigorously? 

Reviewer #1: Yes

4. Have the authors made all data underlying the findings in their manuscript fully available?

Reviewer #1: Yes

5. Is the manuscript presented in an intelligible fashion and written in standard English?

Reviewer #1: No

6. Review Comments to the Author

Reviewer #1: Significant editing is still required. The first part of the title still sounds a little odd. Maybe “Live births from urine derived cells:”. Discussion is long and the flow could be improved. It should be clear earlier in manuscript that for the canine SCNT, cloned donor animals were used.

Line 22-24: Split into two sentences so that it is clear that UDC were used in canine cloning (led to live births) and bovine SCNT (in vitro development only).

Line 29: add comma after 4C

Line 30-32: awkward and unclear. Revise to make it clear that SCNT embryos regardless of donor cell (or species?) did not differ from IVF controls.

Line 33: add apostrophe to species

Line 35: delete comma after material

Line 46: change processes to process

Line 53: cellular based should be hyphenated

Line 55: change life to “live”

Line 56: add “after cloning” after offspring

Line 57: add “s” to application; delete “Offspring from” and add “s” to produce

Line 58: delete for the use in

Line 61: delete s of aspects

Line 65: delete comma after efficiencies

Line 67: add comma after UDCs and before the ability; add SCNT before embryos

Line 68: change induce to “produce”

Line 68-71: you need a transition here to indicate the relevance of mitochondria. It doesn’t fit. Also, many studies have found heteroplasmy in cloned blastocysts and a low percentage in live offspring. A few examples:

- Burgstaller JP, Schinogl P, Dinnyes A, Müller M, Steinborn R. Mitochondrial DNA heteroplasmy in ovine fetuses and sheep cloned by somatic cell nuclear transfer. BMC Dev Biol. 2007 Dec 21;7:141. doi: 10.1186/1471-213X-7-141. PMID: 18154666; PMCID: PMC2323970.

- Takeda K, Tasai M, Iwamoto M, Akita T, Tagami T, Nirasawa K, Hanada H, Onishi A. Transmission of mitochondrial DNA in pigs and progeny derived from nuclear transfer of Meishan pig fibroblast cells. Mol Reprod Dev. 2006 Mar;73(3):306-12. doi: 10.1002/mrd.20403. PMID: 16245357.

- Inoue K, Ogonuki N, Yamamoto Y, Takano K, Miki H, Mochida K, Ogura A. Tissue-specific distribution of donor mitochondrial DNA in cloned mice produced by somatic cell nuclear transfer. Genesis. 2004 Jun;39(2):79-83. doi: 10.1002/gene.20029. PMID: 15170692.

Line 71: change Together these tools to “SCNT may prove” since you are discussing SCNT technology in your introduction.

Line 74: add a comma after “In this manuscript” change to to “into”

Line 80: change to “as a model for human disease”

Line 83: change of to “for” so that it reads “several limitations for culture of canine oocytes…”

Line 91” change cells to “cell”

Line 97: change and to “as they were” so it reads “predominantly in canine samples as they were almost absent from bovine samples.

Line 99: change cells to “cell”

Line 102-106: sentence is too long, suggest breaking up

Line 106: add comma after 72 hrs

Line 107: delete attempted

Line 188: add semicolon before and comma after however “attachment; however,”

Line 129: add period after ablated

Line 134: delete comma after SCNT

Line 135: add a sentence describing the animals the fibroblasts and UDCs were collected from.

Line 136: delete (SCNT) as it was already used in introduction

Line 138: add comma after one

Line 142: add apostrophe to groups “ groups’ ”

Line 144: This is unclear “72% +/- 21SD continued” revision is needed

Line 147-150: sentence is unclear and awkward, revision is needed

Line 154: add comma after transfer

Line 162: change very to “vary”

Line 164: delete are; delete related to and change to “in” so it reads “and appear indistinguishable in their health, vigor and vitality…”

Line 167: change for to “of”; add comma after UDCs

Line 168: change comma to “and”

Line 169: stem cell like should be hyphenated “stem cell-like”

Line 170: add comma after bodies

Line 171: change rational to rationale; clarify what rationale you are referring to here

Line 174: OCT4 should be italicized

Line 178: delete the comma after source

Line 180: hyphenate cell based “cell-based”

Line 182-184: This sentence is unclear. Do you mean to say that it is the first report of using UDCs from a cloned animal to make other clones?

Line 191: delete comma after urine

Line 192 and 193: delete (UDCs) because this abbreviation as already been used earlier.

Line 197: add commas “, also to our knowledge”

Line 198: add semicolon and delete to our knowledge so that it reads “UDCs; the only other report of cloning using UDCs…”

Line 201: delete s on modifications. This is unclear do you mean the methyltransferases that add H3K27me3 or genes regulated by this modification?

Line 202-203: This sentence is unclear and needs revision.

Line 213: delete include

Line 215: delete from

Line 215-220: discussion of the markers does not add much here. What’s the point?

Line 230: what stem cell component? Please elaborate. Maybe belongs with what you are trying to say at line 205??

Line 239: where multiple breeds used? This is not outlined earlier and should be clarified in results. Only Saluki was mentioned in the methods and no info about bovine donor animals is present. Please correct in methods.

Line 244: Figure X needs to be corrected

Line 253-257: This sentence is long and awkward and needs revision.

Line 264: delete and

Line 288-292: resveratrol’s use in culture was not reported in results. Please check and revise

Line 307-308: change to resembled embryoid bodies; delete were observed.

Line 315: delete methods

Line 337-338: Sentence fragment. Revise.

Line 340: delete comma after expansion

Figure 2: no explanation of panel D in legend.

7. PLOS authors have the option to publish the peer review history of their article (what does this mean?). If published, this will include your full peer review and any attached files.

Reviewer #1: No

---

## [Author Response · Author response to Decision Letter 1]

6 Oct 2022

There were a few things which would have been nice to discuss related to the comments on the previous decision letter. Hopefully things have been clarified, but exact formatting requirements compared to those listed would be nice to have addressed, additionally a template would facilitate this for future submissions. 

Thank you for your time looking forward to finalizing this as it has taken a very long time. 

Best. 

- Olof

---

## [Decision Letter · Decision Letter 2]

21 Nov 2022

Live births from urine derived cells

PONE-D-22-06002R2

Dear Dr. Olsson,

We’re pleased to inform you that your manuscript has been judged scientifically suitable for publication and will be formally accepted for publication once it meets all outstanding technical requirements.

Kind regards,

Wilfried A. Kues, Ph.D.

Academic Editor

PLOS ONE

Additional Editor Comments (optional):

Reviewers' comments:

Reviewer's Responses to Questions

**Comments to the Author**

1. If the authors have adequately addressed your comments raised in a previous round of review and you feel that this manuscript is now acceptable for publication, you may indicate that here to bypass the “Comments to the Author” section, enter your conflict of interest statement in the “Confidential to Editor” section, and submit your "Accept" recommendation.

Reviewer #1: All comments have been addressed

2. Is the manuscript technically sound, and do the data support the conclusions?

Reviewer #1: Yes

3. Has the statistical analysis been performed appropriately and rigorously? 

Reviewer #1: Yes

4. Have the authors made all data underlying the findings in their manuscript fully available?

Reviewer #1: Yes

5. Is the manuscript presented in an intelligible fashion and written in standard English?

Reviewer #1: Yes

6. Review Comments to the Author

Reviewer #1: Authors have addressed all of my comments. Another proofread by the authors would catch some typos like two periods. Thank you.

7. PLOS authors have the option to publish the peer review history of their article (what does this mean?). If published, this will include your full peer review and any attached files.

Reviewer #1: No

---

## [Editor Report · Acceptance letter]

13 Jan 2023

PONE-D-22-06002R2 

Live births from urine derived cells  

Dear Dr. WS.:

I'm pleased to inform you that your manuscript has been deemed suitable for publication in PLOS ONE. Congratulations! Your manuscript is now with our production department. 

Kind regards, 

on behalf of

Dr. Wilfried A. Kues 

Academic Editor

PLOS ONE